# Quality and Dosimetric Accuracy of Linac-Based Single-Isocenter Treatment Plans for Four to Eighteen Brain Metastases

**DOI:** 10.3390/cancers17233776

**Published:** 2025-11-26

**Authors:** Anna L. Petoukhova, Stephanie L. C. Bogers, Jeroen A. Crouzen, Marc de Goede, Wilhelmus J. van der Star, Lia Versluis, Masomah Hashimzadah, Jaap D. Zindler

**Affiliations:** 1Department of Medical Physics, Haaglanden Medical Centre, Burgemeester Banninglaan 1, 2262 BA Leidschendam, The Netherlands; l.bogers@haaglandenmc.nl (S.L.C.B.); m.de.goede@haaglandenmc.nl (M.d.G.); j.van.der.star@haaglandenmc.nl (W.J.v.d.S.); 2Department of Radiation Oncology, Haaglanden Medical Centre, Burgemeester Banninglaan 1, 2262 BA Leidschendam, The Netherlands; j.crouzen@haaglandenmc.nl (J.A.C.); l.versluis@haaglandenmc.nl (L.V.); m.hashimzadah@haaglandenmc.nl (M.H.); j.zindler@haaglandenmc.nl (J.D.Z.); 3Department of Radiation Oncology, HollandPTC, Huismansingel 4, 2629 JH Delft, The Netherlands

**Keywords:** multi brain metastases, single-isocenter multi-target, quality control, dosimetric accuracy, LINAC-based

## Abstract

Stereotactic radiotherapy (SRT) is a promising option for treating multiple brain metastases, where minimizing dose to healthy tissue is key to avoiding complications like radionecrosis. Using a single isocenter for multiple targets, rather than a separate isocenter for each lesion, can significantly shorten treatment time. This study evaluates the accuracy of single-isocenter multi-target SRT by comparing the calculated dose in the treatment planning system (TPS) with both film dosimetry measurements and secondary dose calculations. Fifty patients with 4–18 metastases (median = 6; total = 356) were treated using non-coplanar linear accelerator (LINAC)-based volumetric modulated arc therapy. The mean global gamma passing rate was 94.5 ± 4.6% for film measurements and 98.2 ± 1.2% for secondary calculations (3%/1 mm). The results demonstrate good agreement between the TPS, film measurements, and independent dose calculations, supporting the dosimetric accuracy of single-isocenter multi-target SRT for treating multiple BMs.

## 1. Introduction

Brain metastases (BMs) are the most common type of intracranial tumors [1]. Stereotactic radiotherapy (SRT) is currently the preferred treatment method for solitary and multiple BMs over whole brain radiotherapy (WBRT) due to better preservation of neurological function, similar survival rates, and improved quality of life in this palliative setting [2,3,4]. The shift from WBRT to SRT for multiple BMs has since been encouraged by the development of novel radiotherapy treatment techniques, such as single-isocenter multi-target (SIMT) SRT. In SIMT SRT, the isocenter is placed in the geometrical center of all BMs rather than using multiple isocenters for each individual BM (single-isocenter single-target; SIST). For SIMT SRT, a linear accelerator (LINAC) can be used, applying volumetric modulated arc therapy (VMAT) or dynamic conformal arc therapy (DCAT) [5,6,7,8,9,10].

SRT for more than four BMs was initially impractical due to the prohibitively long treatment times. The main advantage of SIMT SRT over SIST SRT is the decreased beam time while preserving plan quality with favorable conformity and dose gradient indices [11,12,13,14,15]. The overall treatment time is significantly reduced by avoiding the need for repeated patient repositioning and imaging sessions for each isocenter [5,16,17]. This makes SRT for patients with more than four BMs feasible in daily clinical practice. A disadvantage of SIMT SRT compared to SIST SRT is the potentially worsened target coverage as a result of rotational errors, particularly for small and distant targets [18,19,20,21,22]. This can generally be mitigated by utilizing a gross tumor volume (GTV)-to-planning target volume (PTV) margin of 1 mm [23,24,25].

The quality of SRT-based plans require a high degree of conformity and a high dose gradient to minimize the damage to surrounding normal tissue [26]. Aside from plan quality, the dosimetric accuracy of SIMT LINAC-based treatments is an important factor. Due to the small volumes of the individual BMs and the potential spread of the BMs throughout the brain, a patient-specific quality assurance (QA) method is required to perform accurate QA, as film dosimetry and knowledge about calibration methods, calibration curves, and dose conversion methods are essential [27,28,29,30].

The aim of this work was to compare the calculated dose in the treatment planning system with the measured dose by film dosimetry and the secondary dose calculation by Mobius3D. For these purposes, SIMT SRT plans of fifty patients with four to eighteen BMs were used to find dependences on the total PTV and number of BMs.

## 2. Materials and Methods

### 2.1. Patient Selection

Patients with multiple BMs were referred for SRT if they had a Karnofsky performance score higher than 60 and a prognosis of more than 6 months. Between August 2021 and August 2023, 50 subsequent patients treated at our center with between 4 and 18 BMs (median = 6, in total 356 BMs) were selected for this study. The exclusion criterion was the presence of any contraindications for MR imaging. There were no exclusion criteria based on lesion location and distribution. Patients who had undergone a neurosurgical resection shortly before SRT were still included in the analysis, with the resected metastases considered in the evaluation. Anonymized data of these patients were utilized (Medical Ethics Committee, reference number N23.034).

Each patient underwent a planning CT scan (Philips Brilliance Big Bore, Eindhoven, The Netherlands) with a slice thickness of 1 mm, using a thermoplastic mask (Q-fix, Avondale, USA) and a bite block to minimize rotational errors. In addition, 3D cerebral MRI scans (Siemens, Den Haag, The Netherlands), including T1-weighted imaging (voxel size 0.9 × 0.9 × 0.9 mm^3^), T2-weighted imaging, diffusion-weighted imaging, and MR perfusion imaging, were also acquired. The Gross Tumor Volumes (GTVs) were outlined by an experienced radiation oncologist, based on areas of contrast enhancement in the T1-weighted MRI scans. This sequence was further utilized for atlas-based segmentation of Organs at Risk (OARs) using RayStation software (RaySearch Laboratories AB, version 10B, Stockholm, Sweden), and the segmentations were subsequently refined by the radiation oncologist, following the European Particle Therapy Network (EPTN) consensus atlas for neuro-oncology [31]. A 1 mm margin was applied from the GTV to the Planning Target Volume (PTV) for all brain metastases (BMs), except those located in the brainstem, where a 0 mm PTV margin was applied due to medical decision. Treatment position verification was carried out using online Cone Beam CT (CBCT), and corrections for translations and rotations were made with a 6D Hexapod couch (Elekta, Stockholm, Sweden) [32,33]. The 6D couch correction was performed both prior to and during treatment, with an additional CBCT scan taken after the correction if necessary. For position verification, an action threshold of 1 mm (vector) and 1° was used, with the action level being applied during CBCT scans with the couch at a 0° angle. When the threshold was exceeded, patients were repositioned. The prescribed dose was 21 or 18 Gy in a single fraction and 25.5 or 24 Gy in 3 fractions. The prescribed dose depended on the PTV of the largest metastasis and was the same for all metastases, according to Zindler et al. [34]. The total PTV volume ranged from 0.56 to 124.5 cm^3^ (median = 7.8 cm^3^). For each PTV, the PTV coverage was at least 99%.

### 2.2. Treatment Planning

Non-coplanar LINAC-based treatment planning was performed in RayStation (RaySearch laboratories, version 10B, Stockholm, Sweden) with 6 MV flattening filter free (FFF, dose rate = 1500 monitor units (MUs)/min) of a Versa HD LINAC (Elekta, Stockholm, Sweden). Versa HD equipped with an integrated Agility multi-leaf collimator (MLC), consisting of 160 interdigitating leaves with a width of 5 mm, was modeled in RayStation [35]. The accuracy of Agility MLC calibration was 0.3 ± 0.17 mm. The isocenter accuracy for non-coplanar arc delivery was 0.25 ± 0.59 mm including couch rotations. The single-isocenter treatment technique consisted of 6 VMAT arcs with 3 couch rotations (0°, 60° and 300°). The collapsed Cone algorithm was used for dose calculations with a dose grid of 1 mm.

The Paddick conformity index (CI), the Paddick gradient index (GI), the total V12Gy and V5Gy of uninvolved brain, i.e., brain volume—GTVs (low V12Gy and V5Gy correlate with favorable plan quality), the number of Mus, and the irradiation time (beam on time) were measured and reported as mean ± 1SD [26,36,37].

Paddick CI was extended for SIMT with multiple targets as shown in Equation (1):(1)$$CI=\frac{V{\left[\left(T_{1}\cap PI_{1}\right)\cup \left(T_{2}\cap {PI}_{2}\right)\cup \ldots \right]}^{2}}{V\left[T_{1}\cup T_{2}\cup \ldots \right]\times V\left[{PI}_{1}\cup {PI}_{2}\cup \ldots \right]},$$
where the treatment target is $T_{i}$ and the prescription isodose is $PI_{i}$ for the ith target, with intersection ∩, union ∪, and structure volumes V[…].

### 2.3. Film Dosimetry and Secondary Dose Calculation

Absolute film dosimetry was conducted using GafChromic EBT3 films (International Specialty Products, USA). These 8 × 10 inch films were handled according to the manufacturer’s recommendations and in line with the American Association of Physicists in Medicine (AAPM) Task Group Report 235 guidelines [30,38]. The EBT3 films have a dynamic dose range of up to 20 Gy, as specified by the manufacturer. After irradiation, the films were left to rest for 24 h before being scanned. The scanning was performed in the same orientation using an EPSON V750 PRO color scanner in 48-bit RGB mode, and the images were analyzed using DoseLab 4.11 [39,40]. Scans were performed in transmission mode, with all color correction and image enhancement features disabled to ensure that only raw data were obtained without any preprocessing. The spatial resolution of the scans was set to 150 dots per inch. To correct for uniformity, an unexposed film from the same batch was also scanned. The RGB data were then converted to optical density (OD) using the green color channel, as described by Micke et al. [29]. Calibration of the EBT3 films was carried out with 6 MV FFF beams on the Elekta Versa HD accelerator, following the procedure outlined by Childress et al. [39]. Calibration of the EBT3 films was performed in a PTW Universal IMRT (PTW Dosimetry, Freiburg, Germany) verification phantom using an ionization chamber (PTW Semiflex 0.125 cm^3^). Eight fields from 50 to 3000 MU with a size of 3 × 3 cm^2^ were irradiated on EBT3 film (see Figure 1). The optical density was fitted to a third-degree polynomial against ionization chamber-measured doses. For absolute dosimetry, the dose in the Semiflex ionization chamber was calculated for a 10 × 10 cm^2^ field in het PTW phantom in RayStation.

For patient-specific QA, treatment plans were recalculated using a custom-made phantom consisting of four layers of plastic (15 × 15 × 2 cm^3^) (see Figure 1). EBT3 films were positioned between the layers for dosimetric measurement. The phantom was positioned to ensure maximum coverage of the metastases on each film. If needed, film dosimetry was performed two or even three times for each patient to obtain all BMs on the films. Each plan was irradiated on this phantom with the original gantry, collimator, and couch angles. Additionally, for each patient, small pieces (6 × 4 cm^2^) of the same EBT3 film with a dose of 15 Gy and 0 Gy were irradiated to check the calibration. The gamma pass rate was calculated with 3% absolute global dose difference and 1 mm distance-to-agreement criteria, with a threshold of 0% [41]. For the evaluation of film dosimetry quality assurance (film QA), the following cut-offs were used:If the passing rate for the global dose difference of 3% and 1 mm distance-to-agreement was >95%, the plan could be approved by a medical physicist.If the passing rate for a global dose difference of 3% and 1 mm distance-to-agreement was 85–95%, and the dose differences were in the low dose range, then the plan could also be approved. If they were in the high dose range, then we checked whether the dose differences were small (within 5%) or large (5–10%) by calculating a relative percent difference to the global dose. We investigated whether it was possible to find out where this difference came from. It could be verified whether doses of the high dose of 15 Gy and the low dose of 0 Gy were measured. We performed the film dosimetry on a different matched LINAC to confirm the results. If necessary, and in consultation with a radiation oncologist, the plan was adjusted.If the passing rate for a global dose difference of 3% and 1 mm distance-to-agreement was <85%, we investigated whether it was possible to find out where this difference came from (for example, wrong unexposed film or dose calibration used, wrong orientation during scanning). If necessary, and in consultation with a radiation oncologist, the plan was adjusted.

Additionally, dose calculations in Mobius 3D software version 3.1 (Varian, Palo Alto, CA, USA) were performed with the same gamma criteria. Mobius3D used the Collapsed Cone calculation algorithm with a dose grid of 1 mm in each direction. If the pass rate for the global dose difference and 1 mm distance-to-agreement was >95%, the plan could be approved by a medical physicist.

### 2.4. Statistical Analysis

For statistical analysis, IBM SPSS version 29.0.2.0 (IBM Corp, Armonk, NY, USA) was used to calculate univariable and multivariable regression for the Paddick conformity index and gradient index. Pearson correlation coefficients (PCC) were calculated to analyze the correlations between two different sets of studied parameters. A two-sided *p*-value ≤ 0.05 was considered statistically significant.

## 3. Results

The total number of irradiated metastases was 356, with a minimum of 4 and a maximum of 18 metastases (Table 1). The metastases had a mean total PTV of 7.76 cm^3^ (range 0.56–124.5). The median number of irradiated metastases was six (range 4–18). The mean number of MUs necessary for irradiation was 6321 (range 3156–10,544). On average, 9162 MUs (range 7672–10,544) and 4263 MUs (range 3156–5690) were needed to irradiate single- and three-fraction plans, respectively. The mean irradiation time of the plans was 600 s (range 359–780) as calculated by Mobius3D. The total treatment time (including positioning of the patient and CBCT verification) needed per fraction per patient was practically feasible within 30 min. The mean V12Gy and V5Gy of uninvolved brain were 37 ± 69 cm^3^ and 157 ± 270 cm^3^, respectively (Table 1). The mean Paddick CI and GI were 0.70 ± 0.10 and 5.2 ± 1.9, respectively. Figure 2 shows the CI and GI as a function of total volume PTV for one and three fractions separately. Lower CI values were observed in the smaller total PTV volumes. Univariable and multivariable regression analyses were performed to evaluate the associations between independent variables (number of metastases, total volume of PTVs, number of fractions) and the rates of the Paddick CI and GI (Table 2 and Table 3).

All performed measurements were similar to the example treatment plans presented in Figure 3 and Figure 4. In each plan, the high target coverage of all individual targets was achieved while minimizing the dose to the surrounding tissue. Indicated by the vertical and horizontal lines, it is possible to draw profiles through the high and/or low dose regions in the film measurements, which indicate the differences even further. The global gamma passing rate of 3% and 1 mm criteria was 94.6% and 96.5% for the patients with nine and five metastases in Figure 3 and Figure 4, respectively. The agreement within the profiles is shown in Figure 3, while the dose in the peaks differ in Figure 4. The average gamma passing rate results of all 50 patients’ film measurements was 94.5% ± 4.6%. Eighteen patients’ plans had results lower than 95% gamma passing rate: 14 patients with a passing rate of 85–95% and 4 patients with a passing rate of <85%: 84.5% (18 metastases), 83.6% (16 metastases), 83.4% (9 metastases), and 83% (6 metastases). These four patients were irradiated in three fractions and had a PTV of 44, 22, 32, and 24 cm^3^, respectively. Depending on the plan, the differences could be in the low dose area or in the peak of the PTV as indicated in Figure 4. Measurements were sometimes higher than the dose calculations in RayStation and vice versa.

The averaged gamma passing rate over 50 patients of the dose calculations in Mobius3D was 8.2 ± 1.2%, with the same gamma criteria as for film QA. Only one patient had a passing rate of 94.9%, lower than the acceptance criteria of 95%.

The dosimetric accuracy was further investigated by calculating the global gamma passing rate of film QA and Mobius3D (Figure 5). Furthermore, the film QA and Mobius3D were compared to the number of monitor units (Figure 5) or compared to the total volume of the PTV (Figure 6). There was no correlation between the film QA and Mobius3D results. The spread of the results of the film QA was higher compared to the results by calculation in Mobius3D. The film QA and Mobius3D dependence on the number of monitor units showed clearly that the number of monitor units was lower for three fractions than for one fraction. As presented in Figure 6, a statistically significant dependence of the gamma passing rates of film QA on the total PTV volume was observed (PCC = −0.391, *p* = 0.003) for all patients, whereas such dependence was minimal for Mobius3D (PCC = −0.218, *p* = 0.064). The global passing rate of the film QA depended on the number of metastases. The passing rate of the film QA decreased faster with the number of BMs for three fractions than for one fraction.

## 4. Discussion

In this study, the difference between the calculated dose and measured dose was quantified for 50 patients with a minimum of 4 and a maximum of 18 brain metastases. The magnitude of the CI and GI was dependent on the total PTV. Univariable and multivariable regression analyses showed statistically significant associations between the total volume of PTVs and the rate of the CI. According to univariable and multivariable regression analyses, the rate of the GI is statistically associated with the number of metastases and the total PTV. The observed lower CI for small total PTVs indicated that, if there are several smaller volumes (<1 cm^3^), it is more difficult to create a conformal plan with a leaf width of 5 mm of Agility MLC. This effect was demonstrated earlier by Gevaert et al. and Petoukhova et al. but had not been described for 356 BMs [7,17]. Moreover, collimator angle optimization in RayStation prevents open leaves between metastases, so-called island blocking. This could not completely solve this problem, and a higher GI for smaller BMs was measured as a result.

Deviations in the gamma passing rate of the film QA below 95% were observed in 18 of the 50 patients analyzed. The location of the differences can be visualized in relative dose difference images. These deviations are mostly in the GTV or in the dose fall off, and a radiation oncologist determined whether replanning was necessary or if the plan was clinically unacceptable. In the occasion when the film results indicate that a plan is too complex for irradiation, the collimator angle optimization was repeated for a different solution to the problem. According to the AAPM Task Group Report 218, gamma analysis criteria are generally recommended by the scientific community for patient-specific QA: 3%/2 mm [42]. For the stereotactic plans, these settings of 5%/1 mm were recommended by the Netherlands Commission on Radiation Dosimetry (NCS) Report 25 [43]. This suggests that the selected gamma criteria of 3%/1 mm and the 95% cut-off for the gamma passing rate may be too stringent in our study.

The mean delivery time of the LINAC plans with six VMAT arcs of 617 s was substantially faster compared to patients who are irradiated on a GammaKnife [44]. Patients benefit also from a shorter setup time when a single isocenter rather than separate isocenters for each metastasis are used at the cost of more integral dose using a LINAC.

According to the AAPM Task Group Report 235, the lateral response artefact (LRA) of film dosimetry is less pronounced when radiochromic films are scanned in the center of the flatbed scanner and when the green channel is used [45]. To reduce LRA artefacts, EBT3 films were placed in the center of EPSON V750 PRO scanner, and for single channel dosimetry the green channel was used. Additional measurements were performed to estimate the LRA artefacts by placing the pieces (6 × 4 cm^2^) of the film right and left from the central position (see Table 4). The LRA effect of EBT3 films was within 2.4% from 100 to 2500 MU. A previous study showed the response curves and fitting procedure error for the green calibration channel of 1.5% [27]. Marroquin et al. studied the total uncertainty in the measured dose for EBT3 films scanned with Epson Perfection V750 scanner [46]. We adapted the table with our results in combination with the measurements by Marroquin et al. (see Table 5). The total accuracy of EBT3 film dosimetry in our measurements was estimated to be approximately 4% of the prescription dose.

Differences between measurements and dose calculations were found to be more acceptable to the physician in high dose areas than in low dose areas. In the low dose areas, organs-at-risk such as the optical system or brainstem are located, which makes it especially relevant to stay within the dose limits. Although film dosimetry has a high resolution, measuring in the dose fall-off of a metastasis, the measurement and calculation were found to have lower agreement than in the high dose areas. Film dosimetry was found to be a time-consuming procedure. This is due to the fact that a waiting time of 24 h is needed between irradiation and creating the result. Special care should be given while introducing this to the clinic, especially to training staff in all the different steps. New high resolution detector arrays might have an advantage since they can be easier to use, and the results can be obtained faster than those obtained by films. For example, a 2D diode array with a diagonal resolution of 2.47 mm, SRS MapCHECK (Sun Nuclear, Melbourne, USA), consists of 1013 diodes with a 0.48 mm × 0.48 mm cross section in a 7.7 × 7.7 cm^2^. Recent publications about SRS MapCHECK in comparison to GafChromic film dosimetry still show a superior resolution of film dosimetry [47]. On the other hand, Infusino et al. demonstrated the higher mean passing rate for SRS MapCHECK than for EBT3 films for pre-treatment verification of CyberKnife stereotactic radiosurgery [48]. Dunn et al. used Sun Nuclear ArcCHECK and SRS MapCHECK, GafChromic EBT films, machine log files, and Varian portal dosimetry to measure different variations of a single SIMT plan [49]. They recommended evaluating SIMT patient-specific QA results against a cohort of patients with a range of PTV sizes and quantities, which we performed in our study. The LRA effect can be more pronounced for GafChromic films than SRS MapCHECK and relate to possible trade-offs. Further comparisons of 2D detector arrays and film dosimetry are needed.

The dosimetric results of the film QA measurements indicate that film measurements are more sensitive compared to calculations with our secondary dose calculation system, Mobius3D. A wider spread is found in three fractions, with lower doses per fraction and therefore fewer monitor units compared to one fraction, indicating that small differences could not be observed by Mobius3D.

SIMT plans are highly complex plans because of a steep dose fall-off of 6 MV FFF and a high number of BMs (four to eighteen in our study). These factors make more stringent demands on quality assurance. This is the reason why we perform film dosimetry as well as secondary dose calculation for each patient with multiple BMs.

Clinical studies indicate that SIMT SRT is an effective and safe treatment option for BMs in terms of local tumor control and (neuro)toxicity [50,51,52,53,54,55,56]. No increased rate of radionecrosis as a result of radiation-induced damage to healthy brain tissue has been reported. One prospective study did not find a significant decline in neurocognitive function or quality of life after SIMT SRT, while local tumor control was maintained [51]. Another study directly compared SIMT SRT and SIST SRT and found no significant difference in local control (*p*-value 0.71). They also found no association between the distance of the metastasis to the isocenter and local recurrence [53]. Another study from our institution also found no association between the isocenter to tumor distance and local recurrence or radionecrosis [56]. Seventy-five patients with a total of 357 brain metastases irradiated with the same SIMT technique were included. The median survival after SRT was nine months. Local recurrence occurred in 7 patients (9%), and 10 patients (13%) developed radionecrosis. At 18 months, the local recurrence-free survival rate was 89%, and the radionecrosis-free survival rate was 85%. Presently, no randomized controlled trials have been performed comparing SIMT and SIST SRT, and no such trials have been registered in ClinicalTrials.gov.

## 5. Conclusions

The aim of this work was to evaluate the quality of the SIMT plans combined with the dosimetric accuracy of LINAC-based delivery for patients with multiple BMs. Patients with four to eighteen BMs were planned with a single isocenter with an overall treatment time of 620 s on average. Acceptable plan quality was observed with a dependence on the volume of the metastasis: a higher volume increases the CI and decreases the GI. Univariable and multivariable regression analyses showed a statistically significant association between the total PTV volume and the rate of CI. According to both analyses, the GI rate was also statistically associated with the number of metastases and the total PTV. The dosimetric results obtained with the film measurements and secondary dose calculations are in good agreement within our criteria, although they did not show a Pearson correlation. In 18 of the 50 patients, the dose measured with GafChromic EBT3 films deviated beyond the applied criteria of a 3% absolute global dose difference and 1 mm distance-to-agreement when compared with the calculated dose. All deviations, however, remained clinically acceptable. This suggests that the selected gamma criteria of 3%/1 mm and a 95% gamma passing-rate threshold may be overly stringent. Measurements involving multiple fractions (with lower dose per fraction) demonstrated lower mean passing rates compared with single-fraction treatments delivering higher doses. This was likely due to differences in maximum dose between one- and three-fraction treatments for the global gamma dose criteria and/or the larger total PTV associated with three-fraction treatments. Overall, these findings support the accuracy of our method and treatment plans.

## Figures and Tables

**Figure 1 cancers-17-03776-f001:**
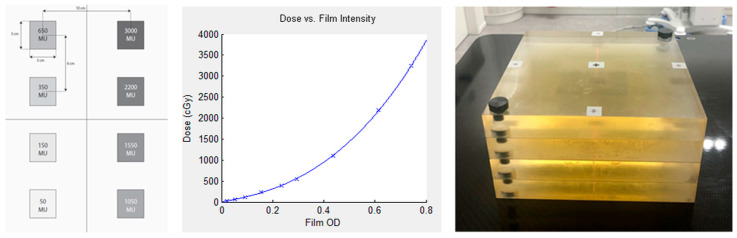
For calibration of EBT3 films, eight fields with a size of 3 × 3 cm^2^ were irradiated with 50 up to 3000 MU (**left**) that resulted in the presented calibration as a function of optical density (OD) (**middle**). For film QA, the home-made phantom consisted of four layers of plastic (15 × 15 × 2 cm^3^) in between the layers of the EBT3 films (**right**).

**Figure 2 cancers-17-03776-f002:**
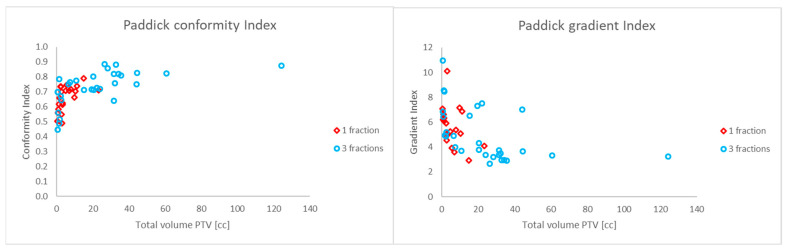
Paddick conformity index (**left**) and Paddick gradient (**right**) of each patient for one (red) or three (blue) fractions.

**Figure 3 cancers-17-03776-f003:**
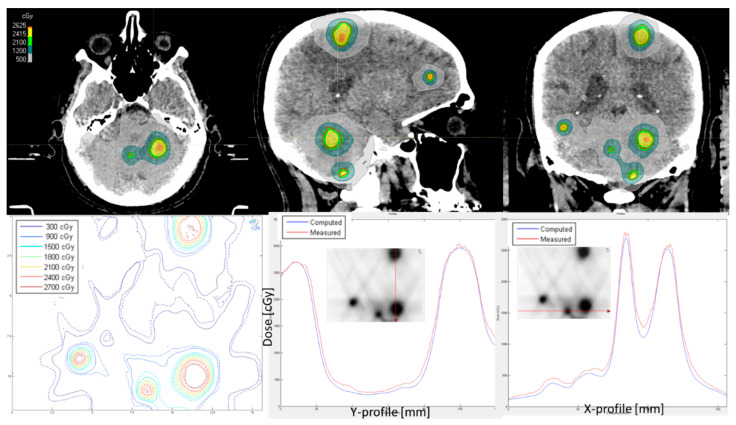
Treatment plan and film QA of a patient with a gamma passing rate of 3% absolute global dose difference and 1 mm distance-to-agreement of 94.6%. The upper row indicates the CT and treatment plan of the patient with nine metastases. The lower row shows isodose lines of the film measurement (dashed) in comparison to dose calculation (solid) followed by a Y- and X-profile through the isodose lines.

**Figure 4 cancers-17-03776-f004:**
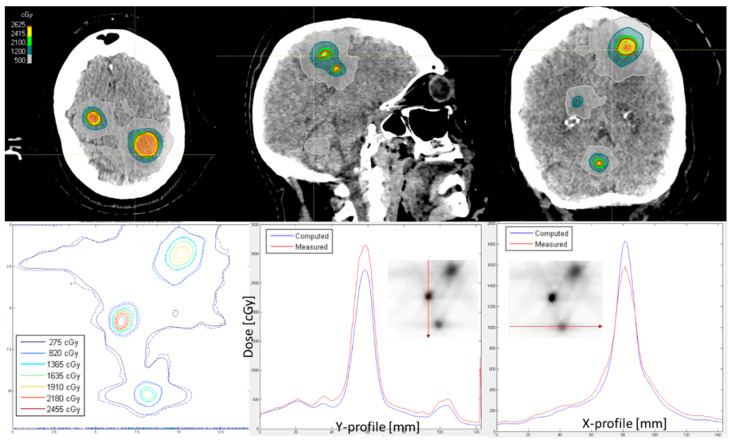
Treatment plan and film QA of a patient with a gamma passing rate of 3% and 1 mm of 96.5%. The upper row indicates the CT and treatment plan of the patient with five metastases. The lower row shows isodose lines of the film measurement (dashed) in comparison to dose calculation (solid) followed by a Y- and X-profile through the isodose lines.

**Figure 5 cancers-17-03776-f005:**
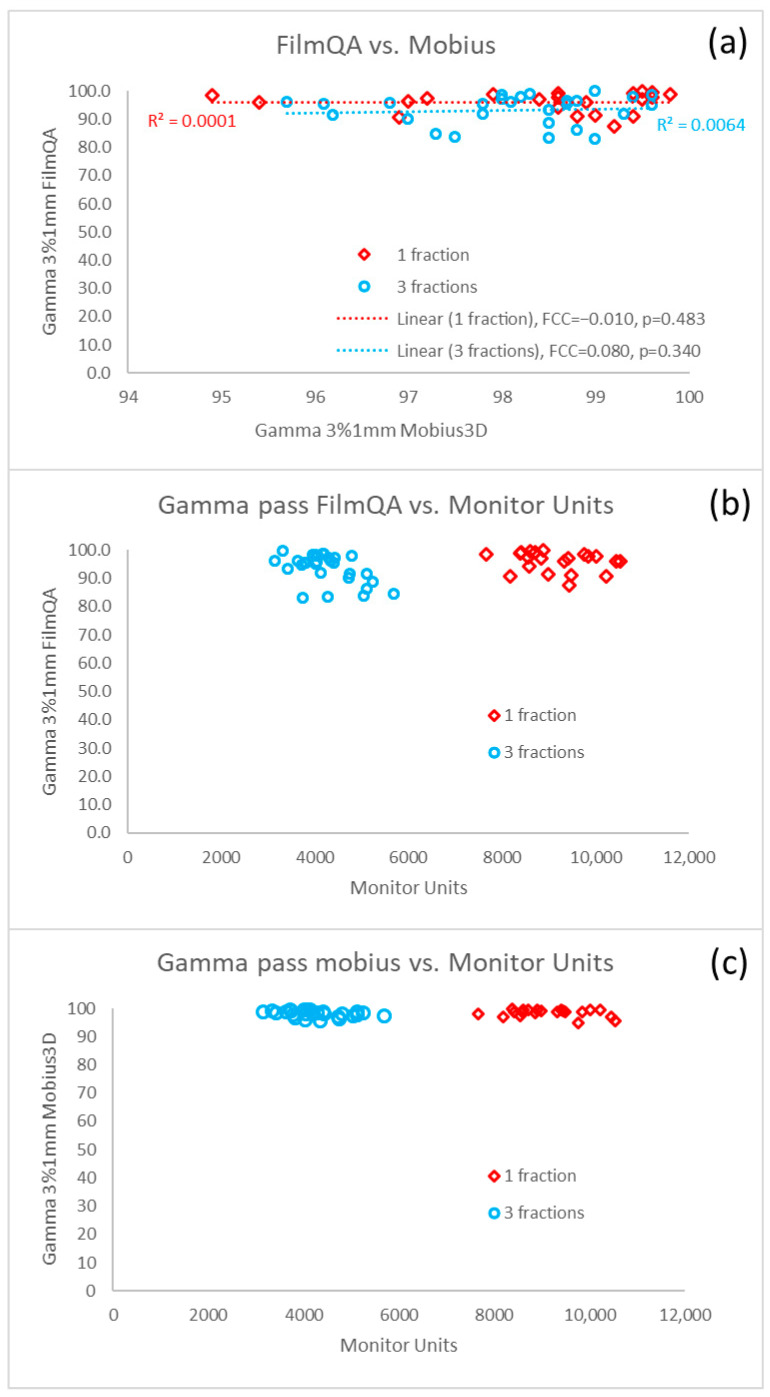
Global gamma passing rate of the film QA and Mobius3D results for one and three fractions (**a**). The dashed lines present trend lines of the results for one and three fractions. For each trend line, Pearson correlation coefficient (PCC) and *p*-value are given in the legend. Film QA results as function of monitor units (**b**). Mobius3D calculation results as a function of monitor units (**c**).

**Figure 6 cancers-17-03776-f006:**
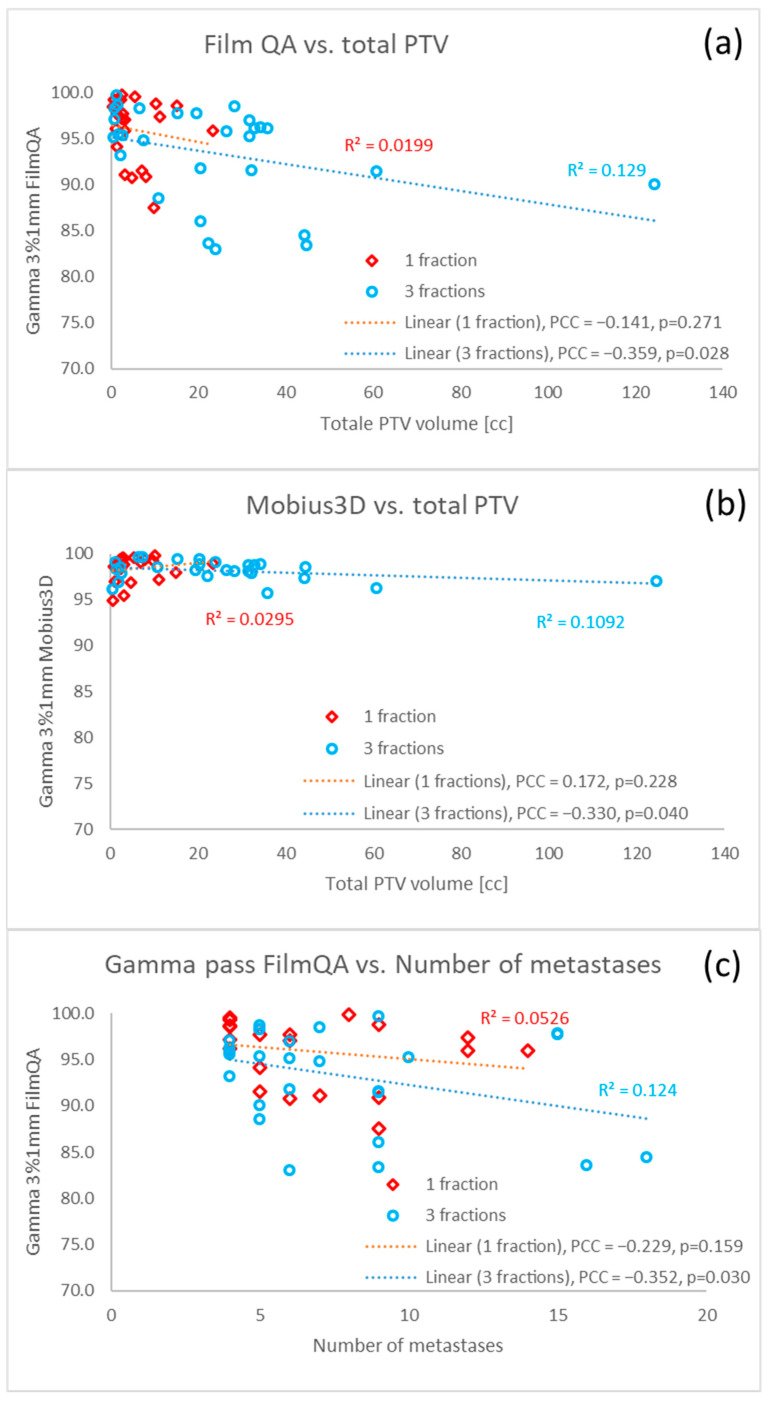
Global gamma passing rates of film QA measurements (**a**) and Mobius3D calculations as a function of total PTV volume for one and three fractions (**b**). Global passing rate of film QA is also presented as a function of the number of brain metastases for one and three fractions (**c**). The dashed lines present trend lines of the results for one and three fractions. For each trend line, Pearson correlation coefficient (PCC) and *p*-value are given in the legend. Note that the y-axis is different to that in Figure 5.

**Table 1 cancers-17-03776-t001:** Patient characteristics.

Patients	Total	50
**Irradiated metastases**	Total	356
**Irradiated metastases**	Median	6
**Irradiated metastases**	Mean	7
**Monitor Units**	Mean ± SD total	6321 ± 2510
	Mean ± SD 1 fraction	9162 ± 765
	Mean ± SD 3 fraction	4263 ± 559
**Total PTV volume**	Median (cm^3^) (range)	7.76 (0.56–124.5)
**Irradiation time [s]**	Mean (s) ± SD	600 ± 90
**Brain-GTV**	V5 (cm^3^) ± SD	157 ± 270
	V12 (cm^3^) ± SD	37 ± 69
**Whole brain-GTVs > 12 Gy [%]**	Mean (%) ± SD	2.5 ± 0.05
**Paddick conformity index**	Mean ± SD	0.7 ± 0.1
**Paddick gradient index**	Mean ± SD	5.2 ± 1.9

**Table 2 cancers-17-03776-t002:** Regression coefficients (B) and *p*-values for Paddick conformity index.

	B (95% CI, *p*-Value)	
	Univariable	Multivariable
**Number of metastases**	−0.001 (−0.01–0.008, 0.83)	-
**Total volume of PTVs**	0.003 (0.002–0.004, <0.001)	0.003 (0.001–0.004, <0.001)
**Number of fractions**		
**1**	1.0 (reference)	1.0 (reference)
**3**	0.071 (0.011–0.13, 0.02)	0.47 (−0.036–0.076, 0.47)

**Table 3 cancers-17-03776-t003:** Regression coefficients (B) and *p*-values for Paddick gradient index.

	B (95% CI, *p*-Value)	
	Univariable	Multivariable
**Number of metastases**	0.194 (0.05–0.337, 0.009)	0.242 (0.12–0.364, <0.001)
**Total volume of PTVs**	−0.04 (−0.064–−0.017, 0.001)	−0.047 (−0.068–−0.027, <0.001)
**Number of fractions**		-
**1**	1.0 (reference)	
**3**	−0.636 (−1.737–0.466, 0.25)	

**Table 4 cancers-17-03776-t004:** Lateral response artefact (LRA) of EBT3 films placed in the central position, 5 cm left, and 5 cm right as a percentage of the central dose for different numbers of MU.

MU	Left (Gy)	Central (Gy)	Right (Gy)	LRA Left	LRA Right
2500	29.82	29.33	29.82	1.67%	1.67%
1750	21.1	20.86	21.37	1.15%	2.44%
1250	15.26	14.93	15.11	2.21%	1.21%
450	5.68	5.58	5.65	1.79%	1.25%
100	1.39	1.39	1.41	0.00%	1.44%

**Table 5 cancers-17-03776-t005:** Summary of dose uncertainties of film dosimetry.

Uncertainties	Green Channel
Response curves and fitting procedure	1.5%
Dose resolution of the system	2.3%
Film reproducibility	0.3%
Film uniformity	0.3%
Lateral response artefact	2.4%
Reproducibility of the response of the scanner	0.3%
Total uncertainty	3.6%

## Data Availability

The original contributions presented in this study are included in the article. Further inquiries can be directed to the corresponding author.

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
