# Peer review of "Quality and Dosimetric Accuracy of Linac-Based Single-Isocenter Treatment Plans for Four to Eighteen Brain Metastases"

_cancers, 2025, doi:10.3390/cancers17233776_

Round 1

Reviewer 1 Report

Comments and Suggestions for Authors

This manuscript presents a well-structured and technically rigorous evaluation of LINAC-based single-isocentre multi-target (SIMT) stereotactic radiotherapy (SRT) for patients with four to eighteen brain metastases. The study uses a large clinical dataset and two independent quality-assurance (QA) verification methods—film dosimetry (GafChromic EBT3) and Mobius3D secondary dose calculations—providing valuable clinical evidence for the dosimetric reliability of SIMT treatments.

Strengths

  • The research addresses a relevant and timely question in modern radiotherapy practice, focusing on the efficiency and accuracy of SIMT SRT.

  • The methodology is well-defined and reproducible, with clear descriptions of patient selection, imaging, planning, and QA protocols.

  • The dual verification approach (film + Mobius3D) and use of strict gamma criteria (3 %/1 mm) strengthen the credibility of the results.

  • The discussion appropriately situates the findings within current literature, demonstrating awareness of TG-235, TG-218, and recent comparative work.

Major Recommendations

  1. Statistical Correlation Analysis – Quantify observed dependencies between total PTV volume, number of lesions, and gamma passing rate (e.g., using Pearson or Spearman coefficients with p-values). This will give statistical weight to statements about correlations.

  2. QA Criteria Justification – The conclusion that 3 %/1 mm, 95 % thresholds may be too strict should be supported by references to AAPM TG-218 or ESTRO QA recommendations, clarifying clinical tolerance levels.

  3. Uncertainty Budget – Expand on the reported 4 % dosimetric uncertainty in film analysis by specifying components (calibration, scanner uniformity, LRA, setup).

  4. Figures 5–6 Enhancement – Add correlation coefficients or confidence intervals to improve data interpretability.

  5. Clinical Context – A brief summary of clinical outcomes (control rate, radionecrosis, or reference to prior institutional experience) would connect technical accuracy to patient benefit.

Minor Recommendations

  • Minor English polishing (e.g., “highly complex plans,” “time-consuming procedure”).

  • Ensure all figure captions fully describe the axes and data points.

  • Confirm consistent formatting of all references to meet Cancers style requirements.

Overall assessment: The study is scientifically sound and clinically relevant. Minor revisions focused on statistical and presentation aspects will make it publication-ready.

Recommendation: Minor Revision.

Reviewer 2 Report

Comments and Suggestions for Authors

The manuscript entitled “Quality and dosimetric accuracy of LINAC-based single-isocentre treatment plans for four to eighteen brain metastases” submitted by Petoukhova et al. evaluates the accuracy of single-isocentre multitarget stereotactic radiotherapy in patients with four to eighteen brain metastases.

The study is very interesting, however, there are some critical points that should be considered:

  1. There is no description of the tumor entity. Were there brain metastasis of lung cancer, breast cancer, colon cancer… Please specify.
  2. There is no subgroup analysis based on the distinct tumor entity. Are there differences based on tumor entity (lung, breast, colon…). Please specify.
  3. There is no description of patient age and sex. Were there differences in the results. Please specify.
  4. There is no subgroup analysis based on number and location of metastasis. This would be interesting. Please specify.
  5. There are no information on side effects. This would be interesting. Please specify.
  6. What about uncertainties regarding measurements and positionings. This would be interesting. Please specify.

Reviewer 3 Report

Comments and Suggestions for Authors

The submitted manuscript by Petoukhova et al represents a robust & methodologically sound evaluation done. They have evaluated the dosimetric accuracy of LINAC-based single-isocentre, multitarget stereotactic radiotherapy in patients with multiple brain metastases. The authors have conducted an in-depth analysis of both GafChromic EBT3 film dosimetry and Mobius3D secondary dose calculations in a population series of 50 patients with 4-18 BMs. The reported global gamma passing rates (94.5% ± 4.6% for film, 98.2% ± 1.2% for Mobius3D with 3%/1 mm criteria) support high agreement between planned and delivered doses.
Detailed analysis identifies a trend for reduced conformity indices and gamma pass rates in smaller or more complex cases, especially with fractionated dosing and larger PTV volumes. While some plans fell below the 95% gamma threshold. But these were considered clinically acceptable ((isn't it?). Well, methodology is robust - carefully balancing meticulous QA c clinical practicality. The study contributes valuable data supporting feasibility & precision of SIMT SRT. I must say that this work is timely & clinically relevant given the increasing utilization of SRT in neuro-oncology.

I have few important suggestions for improvement:

1. Biases in patient selection need to be further explained, especially those related to selection criteria in lesion location and distribution.

2 Strengthen the statistical validation of the correlation, including between the size of the PTV and gamma pass rate, by performing regression or ANOVA.

3. I can see some issues in film dosimetry. Thoroughly discuss implications of lateral response artifacts, and how these relate to possible trade-offs with 2D diode arrays.

4. Can you please include follow-up data on local control or toxicity, when available, to support the dosimetric conclusions with clinical endpoints?

5. Authors must improve clarity of some figures, for example dose difference maps, by possibly adding overlays or quantitative difference summaries.
